# MicroRNA Cross-Involvement in Acne Vulgaris and Hidradenitis Suppurativa: A Literature Review

**DOI:** 10.3390/ijms23063241

**Published:** 2022-03-17

**Authors:** Francesco Borgia, Lucia Peterle, Paolo Custurone, Mario Vaccaro, Giovanni Pioggia, Sebastiano Gangemi

**Affiliations:** 1Department of Clinical and Experimental Medicine, School and Operative Unit of Dermatology, University of Messina, Via Consolare Valeria-Gazzi, 98125 Messina, Italy; ptrlcu92p41a757f@studenti.unime.it (L.P.); cstpla89h16g288z@studenti.unime.it (P.C.); mario.vaccaro@unime.it (M.V.); 2Messina Unit, Institute for Biolomedical Research and Innovation, National Research Council of Italy (IRIB-CNR), Via Vincenzo Leanza, Mortelle, 98164 Messina, Italy; giovanni.pioggia@irib.cnr.it; 3Department of Clinical and Experimental Medicine, School and Operative Unit of Allergy and Clinical Immunology, University of Messina, Via Consolare Valeria-Gazzi, 98125 Messina, Italy; sebastiano.gangemi@unime.it

**Keywords:** hidradenitis suppurativa, acne, mirnas, skin, inflammation, cytokines, biologic therapy, diagnosis, treatment

## Abstract

Acne Vulgaris (AV) and Hidradenitis suppurativa (HS) are common chronic inflammatory skin conditions that affect the follicular units that often coexist or are involved in differential diagnoses. Inflammation in both these diseases may result from shared pathways, which may partially explain their frequent coexistence. MicroRNAs (miRNAs) are a class of endogenous, short, non-protein coding, gene-silencing or promoting RNAs that may promote various inflammatory diseases. This narrative review investigates the current knowledge regarding miRNAs and their link to AV and HS. The aim is to examine the role of these molecules in the pathogenesis of AV and HS and to identify possible common miRNAs that could explain the similar characteristics of these two diseases. Five miRNA (miR-155 miR-223-, miR-21, and miRNA-146a) levels were found to be altered in both HS and AV. These miRNAs are related to pathogenetic aspects common to both pathologies, such as the regulation of the innate immune response, regulation of the Th1/Th17 axis, and fibrosis processes that induce scar formation. This review provides a starting point for further studies aimed at investigating the role of miRNAs in AV and HS for their possible use as diagnostic-therapeutic targets.

## 1. Introduction

Acne Vulgaris (AV) and Hidradenitis suppurativa (HS) are two common chronic inflammatory skin conditions that primally affect the follicular units. Follicular obstruction, dilatation, and rupture are shared mechanisms involved in the pathogenesis of both conditions. Inflammation in these conditions may result from shared pathways, which may partially explain their frequent coexistence [1].

### 1.1. Pathogenesis of AV

AV is the most frequent disease of the skin and is a typical condition, affecting approximately 85% of teenagers [2], with males most frequently affected in younger ages, and females in adult ages [3]. Acne lesions occur mainly in the body areas where the pilosebaceous units are more represented such as the face, back, and chest. The lesions are classically divided into non-inflammatory lesions such as microcomedones, open comedones or blackheads, and closed comedones or whiteheads, as well as inflammatory lesions such as papules, pustules and cysts. In addition, acne scars can result from inflammatory lesions.

There is general agreement that AV has three main triggering factors: hyperseborrhea, growth and proliferation of *Propionibacterium acnes* which, in turn, feeds the hyperproduction of seborrhea and generates follicular inflammation and keratinization of the hair infundibulum with the formation of comedones. The pathogenesis of hyperseborrea in AV is not completely understood, as it seems that it is caused by a major activation of circulating androgen in the pilosebaceous unit due to increased 5-reductase levels and increased expression of androgen receptors in this site.

AV is characterized by an alteration in the lipidic components of the sebum [4]. This alteration acts as a stimulus for the hyperkeratinization of the infundibulum and the subsequent formation of comedones. In addition, excess and change in the composition of the sebum represent an excellent environment for *P. acnes* proliferation. In the hair follicle, *P. acnes* stimulates Langerhans cells via the TLR-2 receptor, which activates Th1 cells by releasing IL-12 and IL-8. *P. acnes* also stimulates the TLR-2 receptor of infundibular keratinocytes, resulting in the overproduction of IL-6 and IL-8, which contribute to the appearance of papules and pustules. *P. acnes* activates the Nod-like receptor 3 (NLRP3) inflammasome in monocytic cells, leading to the production of IL-1β and the increase in Th17 cells and cytokines [5].

### 1.2. Pathogenesis of HS

HS, also known as acne inversa, is less prevalent than AV, and affects approximately 1% of the population. Typically, HS appears in young adults, and females are more affected than males. Clinical aspects of HS are deep-seated painful nodules, abscesses, suppurative sinus tracts or tunnels, bridged scars, and double- and multi-ended comedones localized in the apocrine gland-bearing areas of the body, most commonly, the axillary, inguinal, and ano-genital regions [6]. As with AV, the pathogenesis of HS involves multiple factors. The primary physiopathogenic mechanism in HS consists in the occlusion and consequent inflammation of the hair follicle combined with the dysregulation of the innate and adaptive immune response. Follicular occlusion causes dilatation and subsequent rupture of the hair follicle, with leakage into the surrounding dermis of the follicular content of keratin and bacteria. Staphylococcus species are the most frequently isolated bacteria in HS lesions in addition to an anaerobic polymicrobial microflora [7]. The proliferation of staphylococci determines the recruitment of neutrophils and lymphocytes at the perifollicular level, leading to the formation of abscesses and sinus tracts with consequent architectural alteration of the pilosebaceous unit. The immunopathogenesis of hidradenitis suppurativa involves various cytokines both of a pro-inflammatory and anti-inflammatory nature. The rupture of the follicle and the consequent release of the follicular content seem to activate the NLRP3 inflammasome and caspase-1, with the consequent production of IL-1 β. IL-1β may play a role in hair follicle disruption as it induces the production of metalloproteinases [8]. IL-1 β also induces the production of chemokines which attract neutrophils to the site, triggering the release of other proinflammatory cytokines. The increase in these cytokines is also induced by macrophages and dendritic cells following the release of keratin and other debris that are recognized by the TLR toll-like receptor [9]. Among proinflammatory cytokines, TNF-α plays a prominent role in HS and its levels correlate with disease severity. This cytokine induces the expression of a large number of chemokines which determine the infiltration of various immune cells and the differentiation of lymphocyte clones into Th1 and Th17 subsets [7].

### 1.3. MicroRNAs Biogenesis and Skin Function

miRNAs are a class of endogenous, short (19–23 nucleotides in length), non-protein coding, gene-silencing or promoting RNAs, which regulate the expression of genes via translational repression or degradation of target messenger RNAs (mRNAs). miRNAs match the 3′-untranslated region (3′-UTR) of target mRNAs through sequence complementarity and recruitment of nucleases to repress mRNA expression. High complementarity between target mRNAs and miRNAs causes mRNAs degradation, while partial complementarity prevents the translation of the target mRNAs [10]. The biogenesis of miRNAs starts at the nucleus, where in most cases RNA polymerase II generates long primary miRNA transcripts (pri-miRNAs).

In the nucleus, pri-miRNA are then processed in precursor miRNAs (pre-miRNAs) by the Drosha complex, which is composed of RNase III, Drosha, and the double-stranded RNA (dsRNA)-binding protein, DiGeorge syndrome critical region 8 (DGCR8), and other associated proteins. Pre-miRNAs are then exported by Exportin-5 to the cytoplasm where they are processed by Dicer, a RNase III, in a 21–24 nucletoide-long miRNA duplex. The miRNA duplex is then loaded into the AGO protein and the passenger strand of the miRNA duplex is secreted to produce mature miRNA. The AGO protein then promotes the assembly of a ribonucleoprotein complex RISC, which takes part in the recognition of the targeted mRNAs [11].

MicroRNAs are involved as regulators in various cellular activities including cell growth, differentiation, apoptosis and migration. miRNAs are present not only in the intracellular space, but also in extracellular spaces such as serum, urine, and saliva.

These findings have made miRNAs potential biomarkers for the diagnosis of various diseases. In fact, interest in the role of miRNAs in the pathogenesis of various diseases has increased enormously, also for cutaneous conditions. The high expression of various miRNAs is necessary for normal skin development. The role of miRNAs also seems to be relevant for the correct fixation of the hair in the follicle [12].

At the same time, knowledge regarding the possible involvement of miRNAs also in the pathogenesis of skin diseases is expanding enormously. The role and alteration of miRNAs also influences skin diseases, most prominently in melanoma [13]. Recently, the role of miRNAs has also been reviewed for the pathogenesis of various inflammatory skin conditions such as psoriasis, eczema, atopic dermatitis and toxic epidermal necrolysis [14].

The aim of this study is to review the literature regarding AV, HS and miRNAs involved in the pathogenesis of these conditions and to identify the miRNAs involved in both these diseases in order to explain their similar features, and to recommend possible miRNA applications for diagnostic/therapeutic purposes.

This review was accomplished by searching the online database PubMed. We searched for English articles using these two search strategies: (“mirnas” (all fields) AND “hidradenitis suppurativa” (all Fields)), (“mirnas” (all fields) AND “acne inversa” (all fields)), (“mirnas” (all fields) AND “acne” (all fields)) and (“mirnas” (all fields) AND “acne vulgaris” (all fields)). After the searches, given the great variety of topics and results, a systematic review or a meta-analysis were considered inappropriate due to the low number of articles. As an alternative, we opted for a narrative review.

## 2. MiRNAs in AV and HS: Pathogenetic Role and Therapeutic Strategies

Inflammation is a key feature in the pathogenesis of both AV and HS, with various chemokines and cytokines that contribute to fuel a vicious cycle. However, the nature and causes of this process are not completely understood. The results of this literature review suggest that miRNAs, which regulate inflammation and the innate and adaptive immune responses, may play an important role and could explain some common clinic characteristics of these two conditions. MiR-155, miR-223, miR-21, and miR-146a levels have been found to be high in both HS and AV. In addition, miRNAs play a role in keratinocyte maturation and coalition, leading to different results in relation to the lack or overexpression of their levels.

### 2.1. Anti-Inflammatory MiRNAs in AV and HS

AV and HS are two chronic inflammatory skin conditions in which innate immunity plays an important role in the initiation of the inflammatory response [15]. Since their discovery, miRNAs have been known to play a key role in the fine modulation of the innate inflammatory response. In the reviewed works, miR-146 plays an important role in the regulation of inflammatory innate immune responses.

In 2017, Hessam et al. evaluated the expression levels of various inflammation related miRNAs in HS-affected skin samples and perilesional samples in comparison with healthy controls. They demonstrated that, in HS samples, the expression levels of miR-146a-5p were higher than in the healthy controls [16]. It has been demonstrated that miR-146a-5p expression can be induced by the NF-kB-pathway under the activation of the TLS system. miR-146a-5p, in turn, acts like a negative feedback of TNF-α production by suppressing IL-1 receptor-associated kinase 1 (IRAK-1) and TNF receptor-associated factor 6 (TRAF-6) in response to TLR ligands [17]. Similar results were found in AV by Zeng et al., who demonstrated that the *P. acnes* biofilm can induce the expression of miR-146a in keratinocytes. miR-146a in turn, can “fine-tune” the inflammation response by down-regulating the same pathway involving IRAK-1 and TRAF-6 [18].

Dull et al. recently confirmed the mechanism in which the expression of miR-146 is induced by TLR1/2-4 activation also in sebocytes. They assessed the role of this miRNA by transfecting sebocytes with miR-146a mimic or miR-146a inhibitor. They found that higher miR-146a levels lead to a decrease in IL-8, confirming that miR-146 is a negative regulator of inflammation in acne lesions with an impact on the production of inflammatory cytokines and the number of infiltrating immune cells [19]. It has also been proven that miR-146a expression is crucial for the capacity of T regulatory (Treg) cells to suppress Th1-type responses [20], in line with other chronic inflammatory skin diseases. miR-146 overexpression has been found in other chronic inflammatory skin diseases such as atopic dermatitis (AD) and psoriasis (Pso). In a review [14], Mannucci et al. reported that miR-146a was one of the miRNAs most upregulated in the keratinocytes of patients with AD, and that in a miR-146a deficient mouse model, a strong inflammation was produced with an increased accumulation of infiltrating cells in the dermis. In Pso lesions, miR-146a is overexpressed [21], thus negatively regulating inflammation-related factors.

Xia et al. reported another miRNA mechanism that controls TLR signalling in AV directly by targeting 3′ UTR of TLR2 by miR-143. miR-143 reduces the production of the TLR2 protein whose expression is induced by *P. acnes* and consequently, inhibits the TLR2-mediated inflammatory response in keratinocytes. The induction of miR-143 is also consequent to the activation of TLR2 by Staphylococcal LTA, thus demonstrating the immunomodulating role of *S. epidermidis* [22]. miR-338 is another miRNA with potential anti-inflammatory activity. Liu et al. discovered that, in sebocytes, the levels of TNF- α decrease after treatment with miR-338-3p. They found that this effect is due to the downregulation, operated by miR-338-3p on the PI3K/AKT pathway by targeting PREX2a [23].

Another study regarding AV focused on the interaction between miRNAs and circRNAs. In this latter study five circ-RNAs were discovered to be downregulated in severe AV and three of them with miR-145-5p [24]. miR-145-5p is an miRNA characterized by anti-inflammatory functions since one its targets is the mixed-lineage kinase 3 (MLK3), which is involved in the control of the transcription activity of JNK and NF-kB [25].

Lastly, Yang et al. investigated the interaction between miRNAs and long non-coding RNA (lncRNA) [26]. They provided a new finding in the pathogenesis of AV with the discovery that H19, a lncRNA, acts like a sponge with miR-196, which is an miRNA involved in the miR-196a/TLR2/NF-kB axis.

To date, no study has explored the interaction between circRNA or lncRNA and miRNAs and HS, which merits further attention.

### 2.2. Pro Inflammatory MiRNAs in AV and HS

While the miRNAs outlined in Section 2.1 seem to play a central role in the modulation of inflammation, other miRNAs seem to be involved in the early stages of the inflammatory pathways. In detail, Hessam et al. suggested that miR-155 is an inflammation-related miRNA involved in the early stage of HS. In fact, the levels of this miRNA are high not only in lesional samples of HS, but also in perilesional samples [16]. In arthritic diseases, the overexpression of miR-155 is associated with the lower expression of SHIP-1. The latter is an inhibitor of inflammation, which induces an increased production of proinflammatory cytokines, such as IL-1β and IL-6, which are involved in the development of Th17 cells, and TNF-alpha [27]. The Th17 pathway plays a pivotal role in both HS and AV [28,29]. miR-21 is another possible miR involved in Th17 cell responses. miR-21 levels have been found to be high in HS [16] and AV [30]. The inhibition of miR-21 in mice models results in the inhibition of TH17 cell-related cytokines such as IL-17A, IL-17F, IL-21, and IL-22 [31,32]. Thus, miRNA-21 may be a promising therapeutic target in HS patients in order to dampen the inflammatory process.

Although an increased expression of miR-155 has been demonstrated in AV, the role of this miRNA in AV still needs clarifying. Mann et al. demonstrated that, during inflammatory processes, the NF-kB-miR-155 axis coordinates with the NF-kB-miR-146a axis to regulate the intensity and duration of inflammation in a two-step mechanism. In the early phase of the inflammatory response, miR-155 is rapidly upregulated and targets SHIP1 by activating the IKK signalosome leading to an amplification of the inflammatory signal. Later on, miR-146a is gradually upregulated by NF-kB and, as mentioned above, forms a negative feedback loop by targeting IRAK1 and TRAF6, ultimately attenuating the NF-kB activity in the late phase of inflammation [33]. Figure 1 shows the role of miR-146 and miR-155 in the TLR2/NF-kB-pathway. Targeting this axis with miR-155 antagomiR or miR-146 agomiR appears to be a promising therapeutic approach to inhibit the inflammatory state of both these diseases, either in early or late phases.

### 2.3. MiRNAs and Inflammasome in AV and HS

Although AV and HS are considered to be common conditions, they rarely take part, as a cutaneous manifestation, in the development of a condition consisting of pyoderma gangrenosum, acne, pyogenic arthritis (PAPA), whose spectrum includes: PAPA, pyoderma gangrenosum, acne, suppurative hidradenitis (PASH); pyogenic arthritis, pyoderma gangrenosum, acne, suppurative hidradenitis (PAPASH); psoriatic arthritis, pyoderma gangrenosum, acne, suppurative hidradenitis (PsAPASH); pyoderma gangrenosum, acne, suppurative hidradenitis, ankylosing spondylitis (PASS); pyoderma gangrenosum, acne, ulcerative colitis (PAC) and PSTPiP1-associated myeloid-related-protein syndrome (PAMI) [34].

All these autoinflammatory conditions likely share some important molecular overlaps with AV and HS (notably inflammasome activation and IL-1β production). Shen et al. investigated NLRP3 single nucleotide polymorphism (SNP) and the susceptibility to develop acne lesions. NLRP3 is a molecule involved in the formation of inflammasome which in AV is activated by *P. acnes* to produce inflammatory cytokines. They found that the rs10754558 polymorphism of NLRP3 gene is associated with acne susceptibility. Notably, they discovered that this polymorphism affects the binding affinity site of miR-4273 to target NLRP3 mRNA. This thus provides evidence of the genetic contribution of miRNAs in the regulation of AV pathogenesis, and identifies the SNP–miRNA pair, rs10754558-miR-4273, as an indicator of moderate-severe acne risk [35]. Investigating the role of these polymorphisms could help in establishing the risk in subjects which present this polymorphism.

### 2.4. MiRNAs and Fibrosis Process in AV and HS

Although miR-21 is generally known to be an oncogenic miRNA (onco-miR) [36], miR-21 plays an important role in the fibrotic processes of several diseases [37]. This was also confirmed in AV by Ghumra et al. and interestingly not only in the sampled skin but also in sera, where miR-21 and miR-150 levels are higher than in healthy controls [30]. Similarly, in systemic sclerosis and idiopathic pulmonary fibrosis (IPF), miR-21 levels were high in tissue samples and sera. miR-21 seems to elicit epithelial–mesenchymal transition (EMT) and to negatively regulate the Small Mother Against Decapentaplegic protein 7 (SMAD7) on the SMAD2/3 fibrotic pathway. In addition, serum levels of miR-21 in IPF patients were higher in patients with rapidly progressive disease compared to patients with slowly progressive disease and healthy controls [38]. The role of miR-21 in this pathway is shown in Figure 2.

Post acne and HS scars have a negative impact on the quality of life of the patients, even when the disease has been resolved. Various studies on miRNAs highlight that miRNAs potentially play the role of soluble biomarkers for skin diseases such as psoriasis [39]. Evaluating the serum levels of this miRNAs also in HS patients thus seems very promising since measuring the circulating levels of miR-21 and miR-150 could represent a powerful biomarker for the rapid identification of patients who are more prone to developing scars and tunnels.

Levels of miR-223 are high in tissue samples of HS [16] and AV, and Ghumra et al. found that the highest concentration was in patients with mild scarring [30]. Principal targets of miR-223 are ETC2 (epithelial cell transformation-2), SMAD1, FGFR2 (fibroblast growth factor 2 receptor), FGF2 (fibroblast growth factor 2), FOXO3 (Forkhead box O3), SPP1 (osteopontin), MMP16 (matrix metalloproteinase 16), PTEN, BCL2L11, and JAK-STAT signalling pathways. These targets are involved in several diseases characterized by an upregulation of the fibrosis-related process, such as idiopathic pulmonary fibrosis (IPF) [40], systemic sclerosis (SS) and liver [41], renal [42], lung [43] and myocardial fibrosis [44].

In addition, interestingly in Pso, miR-223 expression in peripheral blood mononuclear cells correlates positively with patients’ PASI scores after etanercept treatment [45]. This thus suggests that miR-223 could be a potential biomarker not only for scar predisposition, but also for the early identification of therapy-respondent patients.

### 2.5. MiRNAs and Keratinization

Alteration of the keratinization process is a characteristic pathological mechanism of both AV and HS. Hessam et al. investigated the role of miRNA regulators in HS samples. For the first time, they demonstrated that miRNA regulators Drosha and Dicer and the RISC complex are downregulated in HS samples [46,47]. An interesting result was that the immunohistochemical staining of Dicer and Drosha demonstrated that there is a shift of positive Dicer-stained keratinocytes in the granular layer of epidermis [46]. This leads to the hypothesis that the downregulation of Dicer is reciprocally related with epidermal proliferation, which is a key feature of HS. These findings are also demonstrated by the significant overexpression of Ki-67 in the basal cells of the epidermis where Dicer was downregulated. Similar findings have been demonstrated in primary breast cancer samples, where Ki-67 overexpression was associated with Dicer and Drosha downregulation [48]. Dysregulated keratinization is also a central feature of AV yet, although much needed, there are no studies on the role of miRNA-regulator alteration in AV.

He et al. found that familiar defective expression of NCSTN leads to decreased miR-100-5p expression, which then results in the upregulation of p-AKT to promote keratinocyte proliferation [49]. No study has focused on the role of miR-100-5p in AV, however it has been proven that the AKT signalling pathway also plays a key role in the pathogenesis of AV [50]. It is therefore also worth evaluating the possible relation between miR-105-5p and AKT pathway in AV.

Finally, miR-146a negatively regulates keratinocyte proliferation via the suppression of the FERMT1 gene, a positive regulator of keratinocyte proliferation needed in the formation of normal skin structure [21].

### 2.6. AV Studies

The principal miRNAs associated with AV are reported in Table 1.

### 2.7. HS Studies

The principal miRNAs associated with HS are reported in Table 2.

In conclusion, this review has compared for the first time two conditions, AV and HS, that share common miRNA patterns. Further knowledge regarding the involvement of miRNAs in these pathologies helps in understanding the pathogenetic mechanisms that explain the clinical presentations featured in both AV and HS. Understanding the role of miRNAs in promoting and modulating the inflammation-related pathways provides the basis for further studies regarding the possible use of miRNAs for diagnostic/therapeutic purposes or as biomarkers of response to a given treatment. Our conclusions are summarized in Figure 3.

## Figures and Tables

**Figure 1 ijms-23-03241-f001:**
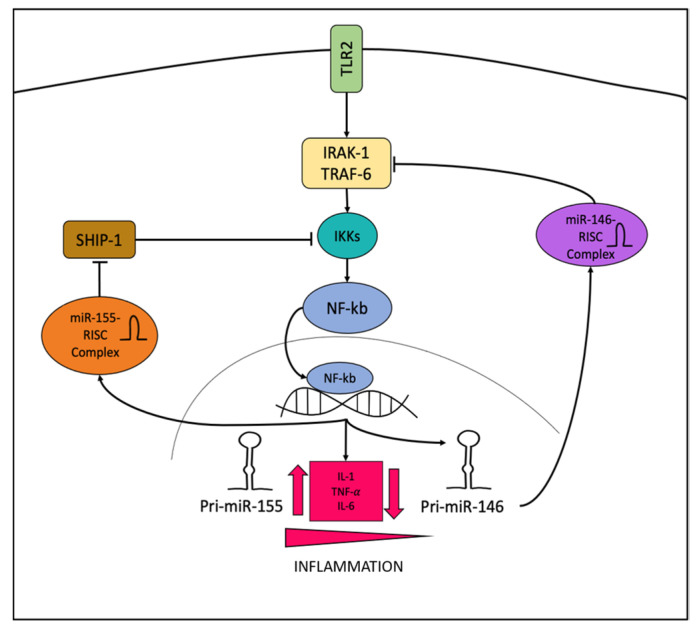
The TLR2/NF-kb pathway: NF-kb, after the activation of TLR2 receptor induces pro-inflammatory cytokines. In addition, it also induces miR-155 which positively promotes the pathway leading to an amplification of the pro-inflammatory signal. On the other hand, miR-146, whose expression is induced later, negatively regulates this pathway, inhibiting IRAK-1/TRAF-6.

**Figure 2 ijms-23-03241-f002:**
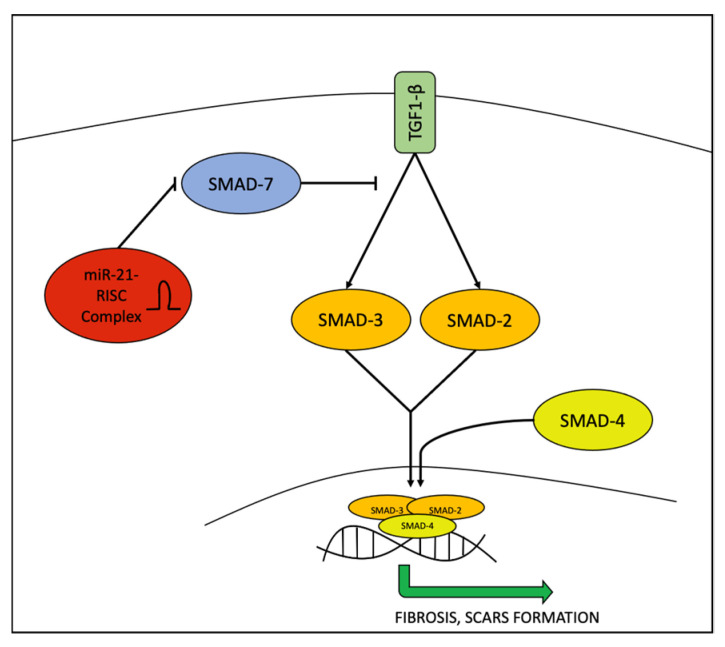
The role of miR-21 in regulating the TGF1-β/SMAD2/3SMAD4 pathway. miR-21 negatively regulates SMAD7, which acts as an inhibitor, leading to the final result of promoting fibrosis process.

**Figure 3 ijms-23-03241-f003:**
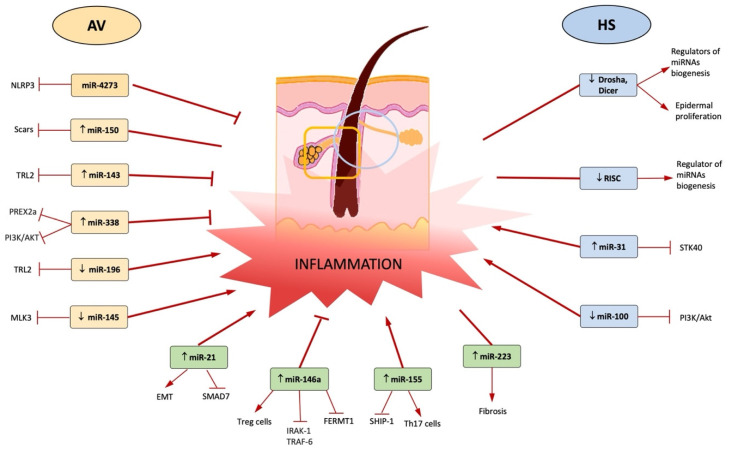
The center of the figure represents the follicular unit, which is the zone affected in both AV and HS. In AV inflammation regards the pilosebaceous unit (yellow rectangle), whereas in HS it regards the terminal hair follicles (blue circle). The miRNAs up- or downregulated in AV are shown on the left, and the miRNAs up- or downregulated in HS are shown on the right. The lower part of the figure shows the miRNAs up- or downregulated in both AV and HS. For each miRNA are reported the target and the final effect on inflammation (red arrows or line) during AV or HS.

**Table 1 ijms-23-03241-t001:** MiRNAs, type of cells, method, design, number of subjects, function, target, year of publication and final effect on inflammation in AV.

miRNAs	Specimen	Method	Design	Subjects n	Function/Target	Year of Publication	Reference
miR-4273	Humans	PCR, dual luciferase reporter assay	Case-control	AV = 428 Controls = 384	Regulation of NLRP3 mRNA expression	2019	[35]
↓ miR-196a	HaCaT cells, human keratinocytes	qPCR, western blotting; ELISA, FISH	Case-control	-	Regulation of TLR2/NF-kb pathway	2020	[26]
↑ miR-146a and miR-146b	Sebocytes, AV and healthy skin biopsy, primary human keratinocytes	miR-146a-5p mimic, miR-146a-5p inhibitor; qPCR, western blotting; ELISA, FISH	Case-control	AV = 10 Controls = 10	Regulation of TLR2/NF-kb pathway, sebocyte activation	2021; 2019	[18,19]
↑ miR-155	AV and healthy skin biopsy, primary human keratinocytes	qPCR	Case-control	AV = 10 Controls = 10	Upregulation of TLR2/NF-kb pathway leading to expression of pro-inflammatory cytokines	2019	[18]
MiR-338-3p	Human sebocytes	Thin-layer chromatography, immunofluorescence, western blotting, flow cytometry	Case-control	-	Targets PREX2 and downregulates PI3K/AKT pathway leading to a reduction of lipogenesis	2017	[23]
↑ miR-223	AV samples, plasma	PCR	Case-control	AV = 8 Controls = 9 41 for circulating miRNAs	High levels in plasma of mild scarring cases, neutrophil activation and resolution of the acute inflammatory response in wound sites	2021	[30]
↑ miR-21	AV samples, plasma	PCR	Case-control	AV = 8 Controls = 9 41 for circulating miRNAs	Inhibits Smad7, a negative regulator of TGF-*β*1/Smad3 signalling	2021	[30]
↑ miR-150	AV samples, plasma	PCR	Case-control	AV = 8 Controls = 9 41 for circulating miRNAs	High levels in plasma of scarring cases	2021	[30]
miR-143	Keratinocytes Tlr2-/- mice, Hela cells, human epidermal keratinocytes	qRT-PCR, ELISA, western blot	Case-control	-	targets 3′ UTR of TLR2 mRNA	2016	[22]
miR-145-5p	AV samples, human keratinocytes	qRT-PCR, Sanger sequencing, PCR	Case-control	AV = 3 Controls = 3	Targets MLK3, a kinase involved in the control of the transcription activity of JNK and NF-kB	2018	[24]

**Table 2 ijms-23-03241-t002:** MiRNAs, type of cells, method, design, number of subjects, function/targets, year of publication and final effect on inflammation in HS.

Type of Cells	miRNAs	Method	Design	Subjects n	Function/Target	Year of Publication	Reference
Keratinocytes from HS samples	↑ miR-31-5p	qPCR	Prospective study	HS = 15 Controls = 10	Enhances expression of proinflammatory mediators in keratinocytes by blocking STK40, a negative regulator of NF-kB signalling	2017	[16]
Keratinocytes from HS samples	↑ miR-21-5p	qPCR	Prospective study	HS = 15 Controls = 10	Maintains T cell-derived skin inflammation, upregulation of TH17 cells and related cytokines, EMT, SMAD 7	2017	[16]
Keratinocytes from HS samples	↑ miR-146a-5p	qPCR	Prospective study	HS = 15 Controls = 10	Regulation of TLR2/NF-kb pathway	2017	[16]
Keratinocytes from HS samples	↑ miR-223-5p	qPCR	Prospective study	HS = 15 Controls = 10	Negatively regulates the proliferation and differentiation of precursor granulocytes into neutrophils	2017	[16]
Keratinocytes from HS samples	↓ RISC	RT-PCR	Prospective study	HS = 18 HS perilesional skin = 10 Controls: 10 healthy controls 10 = psoriatic patients	Regulation of miRNA formation and function	2017	[47]
Keratinocytes from HS samples	↓ Drosha, Dicer, Drosha co-factor DGRC8, Exportin-5	RT-PCR, immunohistochemistry	Prospective pilot study	HS = 18 HS perilesional skin = 7 Controls: 10 healthy controls 10 = psoriatic patients	miRNAs key regulators of biogenesis of miRNAs	2016	[46]
Keratinocytes from HS samples	↓ miR-100-5p	qRT-PCR, in situ hybridization, immunofluorescence, western blotting, cell transfection and cell counting kit-8 assays	Case-control	HS = 5 Controls = 5	Proliferation of keratinocytes targeting AKT pathway	2020	[49]

## Data Availability

Not applicable.

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
