# Peer review of "MicroRNA Cross-Involvement in Acne Vulgaris and Hidradenitis Suppurativa: A Literature Review"

_ijms, 2022, doi:10.3390/ijms23063241_

Round 1

Reviewer 1 Report

The authors have presented an interesting review paper. However, they should present the contents in appropriate paragraphs. The current version includes many paragraphs with only one or two sentences. Good formatting of the article is required.

It is unclear why the authors present a review article in a research article format. The authors should revise the manuscript completely.

Author Response

Dear Reviewer,

thank you for your observations. The paper has been modified accordingly to your suggestions. The article has been formatted differently and presents a form more suitable for a review work. Paragraphs have been modified in order to make more consultable its content. While waiting your new opinions and suggestions, we send

Best regards.

Reviewer 2 Report

This is a good review by the authors on the role of miRNA in HS and acne. It is unclear to me why the authors limited their search to 2016 and after. Due to the limited number of papers reviewed, this manuscript can be improved by adding data from papers prior to 2016. Since there aren't many publications on the topic, I recommend not limiting the search to a specific time frame.  The review can also be improved by addition of a discussion section where the authors can discuss current limitations and future directions.

Author Response

Dear Reviewer,

thank you for your opinions. The review was actually already including results prior to 2016 but, due to a phrase that should have been elided from the final draft, it was referring to hidradenitis, whilst acne includes results since 2012 (proper research). Also, the discussion section has been modified accordingly, fusing it with the results section and adding future perspectives, in order to make a more comfortable reading experience.

While waiting for further notices, we send our

Best regards.

Reviewer 3 Report

In this manuscript for a review article, it was aimed to review the role of microRNAs in acne vulgaris (AV) and hidradenitis suppurativa (HS). The topic of this manuscript would be interesting, but there is limited evidence available in the literature on this topic. Moreover, in general, this paper is poorly written and for most part, it only gives a brief description of the reviewed papers one after the other. In conclusion, this manuscript should be subjected to a very extensive major revision to be deemed for publication in IJMS.

Major remarks:

  1. The Introduction section is very hard to follow as it does not have a clear focus, it includes some facts on AV and HS with many one sentence-long paragraphs which are not always connected. It should be reorganized by focusing on the explanation of the known pathogenesis of AV and HS followed by introducing how microRNAs function.
  2. Further, more complex figures should be included in the manuscript detailing the role of miRNAs in AV and HS. Now there is only one very basic figure, which could have been included as a figure, as it only lists the down- and upregulation of miRNAs leading to inflammation in AV and HS, but pathways are not elucidated.
  3. It is not clear why only articles that were published after 2016 were included in this review.
  4. The list of keywords are very much limited (“mirnas” [all fields] 296 AND “hidradenitis suppurativa” [all Fields]) and (“mirnas”[all fields] AND “acne” [all 297 fields]), these should be extended to other synonims such as “microRNA”, “acne inversa”, etc. Also, the number of identified papers is very low.
  5. The Discussion section does not include an interpretation or comparison of different studies, it just continues with the description of the expression patterns of miRNAs found in certain studies.

Minor remarks

  1. For the tables, the year of publications should be also given. In addition, tables should contain more detailed description of the result of each study, instead of just stating anti or pro-inflammatory effects of miRNAs.
  2. The language of the manuscript is poor in general, it should be carefully reviewed, preferably by a native speaker.

Author Response

Dear Reviewer,

Thank you kindly for your interest. Here’s a point-to-point list with answers:

  1. A) Major

1) The introductive section has been modified as suggested, specifying acne and hidradenitis pathogenesis, and adding how miRNAs work in the skin in general.

2) Two more figures have been added to the paper, explaining current known pathways related to acne and hidradenitis development.

3) Research was conducted also to articles prior to 2016. Due to a typo, the phrase regarding the research extending from 2016 regarded just articles related to hidradenitis when, in fact, articles regarding acne prior to 2016 have already been included.

4) We thank the Reviewer to outline this relevant point: unfortunately, research as suggested (using synonyms such as “acne inversa”) led to no new results. The number of articles is very few indeed but due to the relevance of the argument we suggest further studies are needed. Nonetheless, these terms have been also submitted in the main text (section: “Materials and Methods”).

5) The discussion section has been modified accordingly: in its place, it has been subdivided based on studied miRNA and role, fusing results and observations.

  1. B) Minor

1) Year of publication and targets have been added to the tables, as suggested.

2) Manuscript has been revised for the English language.

Thank you gain for your interest.

Round 2

Reviewer 3 Report

The authors have carried out an extensive major revision, addressing all the raised issues. However, improvement of the English of the paper would be useful.

Author Response

Dear Reviewer,

thank you for your feedback. We are glad the revisions match your expectations. On another note, English has been revised again to make the text more fluid. Hoping these corrections are more suitable for the final product, we send

Best regards.